# Antibacterial Mechanisms and Clinical Impact of Sitafloxacin

**DOI:** 10.3390/ph17111537

**Published:** 2024-11-16

**Authors:** Elian M. A. Kuhn, Levy A. Sominsky, Marco Chittò, Edward M. Schwarz, T. Fintan Moriarty

**Affiliations:** 1AO Research Institute Davos, 7270 Davos, Switzerland; elian.kuhn@aofoundation.org (E.M.A.K.); marco.chitto@aofoundation.org (M.C.); 2Infection Biology, Biozentrum, University of Basel, 4056 Basel, Switzerland; 3Center for Musculoskeletal Research, University of Rochester Medical Center, Rochester, NY 14642, USAedward_schwarz@urmc.rochester.edu (E.M.S.); 4Department of Pathology and Laboratory Medicine, University of Rochester Medical Center, Rochester, NY 14642, USA; 5Medical Scientist Training Program, University of Rochester School of Medicine and Dentistry, Rochester, NY 14642, USA

**Keywords:** sitafloxacin, fluoroquinolone, biofilm, persister, UTI

## Abstract

Sitafloxacin is a 4th generation fluoroquinolone antibiotic with broad activity against a wide range of Gram-negative and Gram-positive bacteria. It is approved in Japan and used to treat pneumonia and urinary tract infections (UTIs) as well as other upper and lower respiratory infections, genitourinary infections, oral infections and otitis media. Compared to other fluoroquinolones, sitafloxacin displays a low minimal inhibitory concentration (MIC) for many bacterial species but also activity against anaerobes, intracellular bacteria, and persisters. Furthermore, it has also shown strong activity against biofilms of *P. aeruginosa* and *S. aureus* in vitro, which was recently validated in vivo with murine models of *S. aureus* implant-associated bone infection. Although limited in scale at present, the published literature supports the further evaluation of sitafloxacin in implant-related infections and other biofilm-related infections. The aim of this review is to summarize the chemical-positioning-based mechanisms, activity, resistance profile, and future clinical potential of sitafloxacin.

## 1. Introduction

Sitafloxacin (DU-6859a) is a broad spectrum, 4th generation fluoroquinolone antibiotic clinically approved in Japan in 2008 and produced under the name Gracevit by Daiichi Sankyo company [1]. Sitafloxacin is utilized against a wide range of Gram-positive and -negative bacterial infections [2]. Most commonly, sitafloxacin is prescribed for pneumonia and urinary tract infections (UTIs); however, it is also approved for other upper and lower respiratory infections (sinusitis, tonsillitis, laryngopharyngitis, and acute bronchitis), genitourinary infections (cervicitis and urethritis), oral infections (periodontitis, pericoronitis, and osteitis of the jaw), and otitis media [1]. It is also a third-line antibiotic for rescue therapy of resistant *Helicobacter pylori* and a second- or third-line treatment against non-gonococcal urethritis caused by both *Mycoplasma genitalium* and *Chlamydia trachomatis* [3,4]. As sitafloxacin is still a relatively new antibiotic, further indications may emerge in the future, particularly since it seems to have potential in infections that are recalcitrant to other treatments. This review explains the mode of action of sitafloxacin, starting with its chemical composition, and summarizes its activity in vitro, in vivo, and in clinical implementation. Finally, the review concludes with potential future applications where the clinical impact and indications for which sitafloxacin may be beneficial are discussed.

## 2. Chemical Composition and Properties

As a 4th generation fluoroquinolone, sitafloxacin has been substantially altered from the initial 1st generation quinolones. The early quinolones originated from the development of nalidixic acid in the 1960s, which is a by-product of the synthesis of the anti-malaria drug chloroquine [5]. These early quinolones evolved into the 2nd generation with the addition of a fluorine and a piperazine ring with norfloxacin and ciprofloxacin being two prominent examples [6,7,8,9,10,11,12,13,14,15]. Subsequently, 3rd and 4th generation fluoroquinolones were introduced in the following decades, which included various additional features such as a broader spectrum of activity (streptococci, anaerobes), longer half-life (enabling once daily dosing) or dual DNA gyrase and topoisomerase IV activity. Prominent examples of these latter generation fluoroquinolones are gemifloxacin, levofloxacin, sparfloxacin and moxifloxacin [16,17,18,19,20,21,22,23,24,25,26]. However, the distinction between the different generations is not clinically important [27].

As is typical for all quinolones, sitafloxacin consists of the quinolone core (quinoline ring fused to pyridone ring) with a keto group (C=O) at position 4 and a carboxylic acid at position 3 (Figure 1). These two groups are essential for binding to the DNA–gyrase complex, which is the target of these antibiotics [27,28]. Small groups like hydrogen (H) are favored at position 2 so as not to interfere with enzyme binding. As a fluoroquinolone, it also has a fluorine at the 6th carbon on the quinoline ring, enhancing antibacterial activity and gyrase potency. In contrast to the 2nd generation fluoroquinolones, the piperazine ring at the 7th carbon in sitafloxacin is replaced with a piperazine derivative (pyrrolidine with an amine and integrated cyclopropane). The addition of this group has contributed toward increased potency against Gram-positive bacteria compared to the original piperazine ring, which enhances activity against Gram-negative bacteria. Similar to some other fluoroquinolones, sitafloxacin also contains an alkyl substitution that contributes to even higher activity against Gram-positive bacteria and increased serum half-life. At the N1 position, it further contains a non-polar cyclopropyl (three-membered carbon ring) group, which was shown to be the best group to improve overall potency and pharmacokinetics. In contrast to other fluoroquinolones, the cyclopropyl in sitafloxacin contains a fluoride substitution (fluorocyclopropyl). At the 8th carbon of the main quinolone ring, sitafloxacin has a chloride substitution, which improves oral uptake and activity against anaerobic bacteria [28]. The fluorine on the cyclopropyl and the chlorine collectively pull the electrons from the neighboring carbon atoms closer. This increases the hydrophobicity of sitafloxacin compared to other fluoroquinolones, which can influence the uptake of sitafloxacin through the bacterial envelope [29,30].

Fluoroquinolones have been shown to have reduced activity at low pH [31], which may be due to changes in charge of the functional groups at different pH values. For example, the carboxylic acid can lose a proton at high pH (-COO^−^), while the amine group can become protonated at low pH (-NH_3_^+^). However, whether sitafloxacin has reduced efficacy at low pH has not been determined to date to the best of our knowledge.

In addition to influencing pH-dependent activity, the carbonyl groups of the main quinolone composition are responsible for the interaction with the DNA gyrase complex within bacteria [32,33,34,35].

## 3. Mechanism of Action

Like all fluoroquinolones, sitafloxacin targets prokaryotic DNA gyrase and topoisomerase IV enzymes. DNA gyrase is responsible for the catalytic relaxation of negatively supercoiled DNA and topoisomerase IV disentangles newly replicated DNA, both of which are essential for transcription and replication [36,37,38,39]. DNA gyrase is composed of two subunits, GyrA and GyrB. Topoisomerase IV is composed of the subunits ParC and ParE. Both protein complexes bind two double strands of DNA and cleave one of them. This allows the second DNA to pass through the gap, after which it is united again (Figure 2). The inhibition of these enzymes by fluoroquinolones is not fully understood, but binding of the fluoroquinolone to both DNA gyrase or topoisomerase IV and DNA is believed to be important [38,40]. All fluoroquinolones block the topology changes of DNA, and the whole complex leads to double-strand breaks, which results in cell death [41,42,43,44,45]. Only the latter generation fluoroquinolones, such as sitafloxacin, can target both enzymes with similar affinity, which is called dual targeting [46].

Sitafloxacin has a balanced inhibition of both enzymes, as revealed by its IC_50_ ratio of 1.4. This compares with other fluoroquinolones, such as moxifloxacin (6.24) and ciprofloxacin (110), which have a greater activity against one of the enzymes. Furthermore, sitafloxacin showed the highest direct inhibitory activity against purified DNA gyrase and topoisomerase IV in vitro compared to levofloxacin, ciprofloxacin, gatifloxacin, tosufloxacin, and sparfloxacin [47,48,49].

## 4. Spectrum of Activity

Sitafloxacin is utilized against a wide range of Gram-positive and -negative bacterial infections, including *Staphylococcus* spp., *Streptococcus pneumoniae*, other *Streptococcus* spp., *Enterococcus* spp., *Moraxella catarrhalis*, *Escherichia coli*, *Citrobacter* spp., *Klebsiella* spp., *Enterobacter* spp., *Serratia* spp., *Proteus* spp., *Morganella morganii*, *Haemophilus influenzae*, *Pseudomonas aeruginosa*, *Legionella pneumophila*, *Peptostreptococcus* spp., *Prevotella* spp., *Porphyromonas* spp., *Fusobacterium* spp., *C. trachomatis*, *Chlamydophila pneumoniae*, and *Mycobacterium pneumoniae* [2]. Sitafloxacin has remarkably low minimal inhibitory concentrations (MICs) compared to other fluoroquinolones for a number of medically important bacterial species. For example, its MIC(90) was 0.06 µg/mL against clinical isolates of *Streptococcus milleri*, which is 8-fold lower than that of ciprofloxacin and 16-fold lower than that of levofloxacin [50,51,52,53,54,55,56,57,58]. Why this is the case is not quite understood. It is not related to the drug accumulation within the bacteria, as sitafloxacin showed very low accumulation compared to other quinolone antibiotics in *S. pneumoniae* [59]. Sitafloxacin also has a lower in vitro MIC against more anaerobic bacteria than older fluoroquinolones such as ciprofloxacin or ofloxacin [60,61,62].

A common factor limiting antibiotic efficacy is related to bacterial metabolic and respiratory activity, which is altered in small colony variants (SCVs) and so-called persisters [63,64,65].

SCVs can be found in patients with chronic Staphylococcal infections, which are in most cases linked to device or prosthetic joint-related infections or osteomyelitis [66,67,68,69]. Antibiotics, including the aminoglycoside gentamicin and the fluoroquinolone moxifloxacin, could not reduce the bacterial numbers of chronic SCVs in vitro and in vivo in an osteomyelitis mouse model. Moxifloxacin was even shown to induce increased SCV formation [70]. However, another study identified sitafloxacin as a potent bactericidal treatment for SCVs of *Staphylococcus aureus* [71]. The study was especially interesting, as sitafloxacin not only killed SCVs of methicillin-susceptible *S. aureus* (MSSA) but also of methicillin-resistant *S. aureus* (MRSA) [71].

Persisters are bacterial sub-populations, which are, amongst other things, more tolerant to antibiotics. They are often connected to recurrent infections and are especially abundant in biofilms [72,73,74]. Sitafloxacin has been proven to kill persisters of *P. aeruginosa* [75]. As *S. aureus* and *P. aeruginosa* can both lead to very problematic infections by forming biofilms [76,77,78,79,80,81], it is especially crucial to note that these studies also saw a potent anti-biofilm activity of sitafloxacin against these two pathogens. Sitafloxacin lead to a 4 log10 reduction in surviving bacteria within *S. aureus* biofilms, which is 100-fold more effective than gentamicin [71]. It also eliminated biofilms of *P. aeruginosa* in vitro by 7 logs and reduced the biofilm survival of an in vivo implant-related infection mouse model by at least 2 logs [75].

Sitafloxacin also displayed greater activity against intracellular *S. aureus* in infected murine macrophages in vitro than levofloxacin or moxifloxacin and showed good antibacterial activity in a mouse peritonitis in vivo model [82]. Fluoroquinolones generally seem to accumulate within monocytes [83]; however, this is not universal within this class of antibiotics, as moxifloxacin does accumulate, but levofloxacin does not [84]. With regard to cytotoxicity, sitafloxacin did not show adverse effects (AEs) against human embryonic kidney 293T cells up to a concentration of 64 µg/mL [71].

## 5. Antibiotic Resistance

As fluoroquinolones have been heavily prescribed in recent decades, resistance is rising [85,86,87,88,89,90,91]. However, few studies have been published so far concerning resistance to sitafloxacin. Resistance has been shown to develop during treatment with 1 out of 10 patients carrying sitafloxacin-resistant *M. genitalium* after treatment failure in one clinical report [4]. Still, resistance to sitafloxacin usually increases the MIC much less in comparison with other fluoroquinolones [46,53]; for example, a point mutation in *gyrA* led to a 16-fold increase in MIC for sparfloxacin or moxifloxacin but only a 4-fold increase in MIC for sitafloxacin [47].

The dual targeting of sitafloxacin against both DNA gyrase and topoisomerase IV might lead to reduced resistance formation, as suggested for the similar fluoroquinolone moxifloxacin [92,93,94,95]. This might be because the most common mechanism for antibiotic resistance in fluoroquinolones is point mutation in the quinolone-resistance-determining regions (QRDRs) of *gyrA* and/or *parC*. In Gram-negative bacteria, weaker resistance has been shown to usually only come from mutations in *gyrA*, while stronger resistance seems to originate from mutations in both *gyrA* and *parC* genes [96,97]. Mutations in *parC* are more likely to be found in Gram-positive bacteria [98,99]. In a combination treatment with doxycycline, sitafloxacin was shown to be more efficient in treating patients infected by *M. genitalium* having a mutation in *parC*. Still, this mutation was found in 50% of cases with failed doxycycline-sitafloxacin treatment, and treatment failure was even more likely if there was also a *gyrA* mutation [100]. Another clinical study checked for mutations prior to initiating sitafloxacin monotherapy against fluoroquinolone-resistant *M. genitalium*. They found a 100% cure rate for patients infected with *M. genitalium* without having any *parC* or *gyrA* mutations, while it was 92.9% for *parC* only mutations and 41.7% for *parC* and *gyrA* double mutants [101].

In *S. pneumoniae*, the mutant prevention concentration (MPC) was much lower for sitafloxacin (1 mg/L) compared to other fluoroquinolones such as ciprofloxacin (128 mg/L), levofloxacin (64 mg/L) and moxifloxacin (8 mg/L) and further showed less mutant frequencies [47]. Alteration of the drug transport mechanism, for example, the upregulation of efflux pump genes, can also lead to resistance in certain cases, specifically for the 2nd generation fluoroquinolones ciprofloxacin and norfloxacin [59,102]. In *S. pneumoniae*, sitafloxacin accumulated the least compared to all other fluoroquinolones tested, which might suggest it is pumped out by efflux systems. However, inhibiting potential efflux pumps did not change the intra-bacterial concentrations of sitafloxacin nor its activity measured in MIC [59]. Therefore, efflux pumps do not seem to be the primary resistance mechanism against sitafloxacin, at least not in *S. pneumoniae*.

Another mechanism for increased fluoroquinolone resistance is downregulation of the target enzyme, such as topoisomerase IV in *S. aureus*, which increases the MIC drastically [103]. Furthermore, the AAC(6′)-Ib-cr mutant protein has been found to be carried on a plasmid in several clinical isolates of Gram-negative bacteria, which makes some fluoroquinolones ineffective. Those enzymes inactivate the piperazine ring and therefore lead to resistance to fluoroquinolones with this chemical composition, such as ciprofloxacin [104]. Sitafloxacin has a piperazine derivative, which means it is not clear yet if the AAC(6′)-Ib-cr mutant protein would also lead to resistance against this antibiotic.

Sitafloxacin has been shown to be highly active against fluoroquinolone-resistant mutants. For example, sitafloxacin remains active against mutated DNA gyrase and topoisomerase IV in resistant *M. genitalium* [105]. Similarly, *H. pylori* with resistance to levofloxacin remained susceptible to sitafloxacin in vitro [106,107]. Furthermore, another study showed a high activity of sitafloxacin against levofloxacin-resistant *E. coli* [108]. Sitafloxacin was also shown to be more effective against *gyrA* mutants of *M. tuberculosis* tested on clinical isolates compared to moxifloxacin, levofloxacin, and ciprofloxacin in vitro [109], making it a valuable option against infections with resistance against commonly used fluoroquinolones.

However, sitafloxacin has been described to display reduced activity against ciprofloxacin-resistant strains [58]. This suggests there can be a shared resistance for more than one of these antibiotics from the same class. For *M. tuberculosis*, a cross-resistance between sitafloxacin and moxifloxacin was observed in 40.6% of the tested clinical isolates in vitro [110]. It seems as if the specific mutation location can either lead to resistance to sitafloxacin or not, so they are not universal between different fluoroquinolones.

Considering the increase in resistance, linked with increased mutation rates in bacterial DNA, the use of fluoroquinolones may warrant careful monitoring into the future. The mechanism of higher mutation rates is not completely clear, but the increased mutation rate is a consequence of DNA repair [27,111,112,113,114,115,116,117,118] and can lead to bacteria adapting to new drugs more quickly.

The resistance mechanism is not universal between classes, as sitafloxacin was shown to be highly effective against multiple bacteria resistant to an antibiotic of another class. One example is the antibacterial activity of multi-resistant enterococci in vitro, where sitafloxacin was effective against strains resistant to antibiotics of any tested class except to fluroquinolones [119]. Another example is the successful eradication of carbapenem-resistant *E. coli* or *P. aeruginosa* and vancomycin-resistant *E. faecium* [120].

## 6. Synergy with Other Drugs or Compounds

Combining sitafloxacin with another antibacterial may increase killing efficacy as well as reduce antibiotic resistance development. Therefore, several studies tested combinations between sitafloxacin and other antibiotics in vitro. For example, combining sitafloxacin with the protein synthesis targeting aminoglycoside antibiotic arbekacin led to a synergistic effect on the majority of the tested clinical isolates of *Mycobacterium abscessus* [121]. Combination with the aminoglycoside amikacin or the cell-wall inhibiting beta-lactam antibiotic imipenem was also beneficial for some of the strains [121]. Sitafloxacin further showed high in vitro activity for extensively drug-resistant *Acinetobacter baumannii* and showed possible synergy with the RNA synthesis inhibiting antibiotic rifampicin and the polymyxin antibiotic colistin [122]. The combination of sitafloxacin with colistin was even able to kill colistin-resistant *A. baumannii*. This could be due to the membrane-disruptive properties of colistin, which would increase uptake of the hydrophobic sitafloxacin [123]. Against *Mycobacterium ulcerans*, the combination of sitafloxacin and rifampicin also led to synergistic effects [124].

Some synergies observed in vitro could also be reproduced in vivo, such as the combination of sitafloxacin and rifampicin against *M. ulcerans* in a mouse footpad infection study [125]. However, synergy between sitafloxacin and colistin for the treatment of carbapenem-resistant *A. baumannii* that was observed in vitro was not found in a patient study compared to colistin monotherapy [126].

Antimicrobial activity can also be increased by partnering with other drugs that influence the drug-inhibiting pathways or antibiotic resistance mechanisms. Adding sitafloxacin together with the beta-lactamase inhibitor sulbactam increases the activity against multiple strains of *A. baumannii* [122,127]. Another approach is to change the environment the drug will be facing to make it more active or simply make the drug reach the target more efficiently. In vitro testing on *H. pylori* showed synergy with lansoprazole, which is a proton pump inhibitor (PPI) used to reduce stomach acid [53]. For both studies, the mechanisms behind these enhanced treatments have not been further investigated.

By conjugating sitafloxacin with a bisphosphonate, the bacterium *S. aureus* can be reached within the canaliculi of the bone shown in a murine model, as the bisphosphonate binds to bone and therefore transports the drug to the bone infection site [128,129,130].

## 7. Utility in Clinical Settings

The primary agent causing UTIs, *E. coli*, is becoming increasingly resistant to established therapeutics. Over 30% of UTI *E. coli* isolates in the Asian–Pacific region were resistant to third and fourth generation cephalosporins. Of these, half also had decreased susceptibility to commonly used fluoroquinolones such as levofloxacin and ciprofloxacin, which are the first-line treatments for complicated UTI [131,132,133]. Sitafloxacin exhibits similar efficacy compared to established levofloxacin therapy (88.6% vs. 94.9%) for uncomplicated UTIs, corroborating its utility as an option to overcome emerging resistance amongst uropathogens [134].

Sitafloxacin also retains its efficacy when used in the treatment of acute pyelonephritis (kidney infection) and complicated UTI, which refers to an infection in specific patient populations (elderly, immunocompromised, pregnant, male) or patients with structural/functional abnormalities of the urinary tract (prostatic hypertrophy, indwelling catheter, neurogenic bladder) [135]. A study in Thailand of 289 patients with complicated UTI deemed sitafloxacin non-inferior to ceftriaxone with clinical success rates of 86.6% vs. 83.8% for intention-to-treat analysis and 97.2% vs. 99% for per-protocol analysis [136]. Another study in China of 59 patients found superior cure rates of sitafloxacin (80%) compared to levofloxacin (54%) for complicated UTI [134].

Regarding other genitourinary infections, sitafloxacin eradicated *M. genitalium* in 92.3% of patients with uterine cervicitis with analogous rates to azithromycin and moxifloxacin [137]. Another study demonstrated sitafloxacin could clear non-gonococcal urethritis in 95.7% of patients with *C. trachomatis*, 93.8% with *M. genitalium*, and 100% with *Ureaplasma urealyticum* [138]. In addition, sitafloxacin is being continuously evaluated for its safety and efficacy in other urologic infections, such as bacterial prostatitis and epididymitis [139]. With the incidence and degree of adverse events remaining similar between sitafloxacin and control groups, sitafloxacin is a valuable addition to the arsenal of antibiotics physicians can use to treat infections of the genitourinary tract.

Sitafloxacin has also been shown to be effective against respiratory infections such as community-acquired pneumonia (CAP). With the most common pathogens including *S. pneumoniae*, *H. influenzae*, and *M. catarrhalis*, CAP has also become more challenging to treat due to increased resistance to empiric therapy, consisting of penicillins, macrolides, and fluoroquinolones [140,141,142]. In a study comprising 300 *S. pneumoniae* clinical isolates in China, 44.7% were classified as resistant to oral penicillin, 60% were resistant to oral cefuroxime, and 96% were resistant to erythromycin; in contrast; all strains were found to be susceptible to levofloxacin [143]. Furthermore, the American Thoracic Society and Infectious Diseases Society of America provide a strong recommendation for fluoroquinolone monotherapy in CAP outpatients with comorbidities such as heart, lung, liver, renal disease, and diabetes mellitus given the class’s broad coverage of CAP-causing organisms, decreased resistance rates, high oral bioavailability, and increased likelihood of patient compliance with a single antimicrobial [142]. Amongst fluoroquinolones, oral sitafloxacin has been ascribed similar cure rates (>92%) for CAP when compared to oral levofloxacin and moxifloxacin with the highest cure rates presenting at a 100 mg bid dosing regimen [144]. Intravenous (IV) sitafloxacin also had comparable cure rates to the beta-lactam antibiotic imipenem (94% vs. 97%) in patients hospitalized with pneumonia [145]. Considering sitafloxacin’s high efficacy against inpatient and outpatient cases of pneumonia and decreased resistance of respiratory pathogens to fluoroquinolones, there is rationale for sitafloxacin use for respiratory infections, especially when first-line therapeutics fail or evoke AEs.

Sitafloxacin is also used as a third-line rescue therapy for *H. pylori* infection, which is highly associated with gastric adenocarcinoma, mucosa-associated lymphoid tissue lymphoma, and peptic ulcer disease [146,147,148]. Increased rates of antibiotic resistance have rendered first-line (PPI–clarithromycin–amoxicillin) and second-line (PPI–metronidazole–amoxicillin) triple therapy less efficacious [149,150]. In Japan, a randomized-control trial (RCT) established that sitafloxacin is superior to levofloxacin, and guidelines recommended its inclusion as part of the third-line regimen for *H. pylori* [150,151]. A systematic review of 12 clinical studies found 7-day treatment with the acid blocker vonoprazan or PPI–sitafloxacin–amoxicillin to have an eradication rate of 80.6% (95% CI, 75.2–85.0) [151,152,153,154,155,156,157,158,159,160,161,162,163]. Importantly, logistic regression analysis in another study demonstrated that decreased susceptibility to sitafloxacin, determined by increased MIC and association with *gyrA* mutation, independently influenced clearance by sitafloxacin-based triple therapy [159]. Therefore, given the duration of treatment and potential for fluoroquinolone-associated AEs, sensitivity to sitafloxacin should be confirmed prior to the initiation of therapy.

## 8. Safety Profile and Adverse Effects

As with most fluoroquinolones, the clinical usage of sitafloxacin requires careful consideration of both the benefits and potential life-threatening risks to the patient. Common AEs of the quinolone class that should be monitored in all patients include but are not limited to gastrointestinal (gastritis and hepatotoxicity), neurologic (altered mental status and peripheral neuropathy), cardiovascular (QT interval prolongation leading to torsades de pointes, a deadly arrhythmia), tendinopathy, dysglycemia, and phototoxicity [164,165,166,167,168,169,170,171,172]. In contrast, commonly reported side effects for sitafloxacin are more limited to the gastrointestinal tract [2]. Sitafloxacin elicits a dose-dependent mild phototoxic effect in Caucasian subjects, which was not observed in Asian subjects [173].

A meta-analysis of four RCTs that compared the safety of sitafloxacin in the treatment of a variety of bacterial infections, mainly UTI and pneumonia, demonstrated a similar risk of AEs between sitafloxacin and comparator antibiotics, including other fluoroquinolones (OR, 1.14; 95% CI, 0.64–2.01; I2 = 61%) [136,145,174,175,176]. Another study evaluating therapeutic efficacy and safety in CAP (*n* = 340) did not find a significant difference in the incidence of drug-related AEs (including clinical, lab, or electrocardiogram abnormalities) between sitafloxacin 100 mg once daily (qd) (29.8%), sitafloxacin 100 mg twice daily (bid) (29.8%), and moxifloxacin 400 mg qd (28.2%). The most common clinical AE in the sitafloxacin 100 mg qd group was dizziness, while that in the 100 mg bid group was nausea/diarrhea. Notably, an increased frequency of sitafloxacin dosing did increase rates of alanine transaminase/aspartate transaminase elevation from 2.6% in the qd group to 9.6% in the bid group, speaking to a potential dose-dependent deleterious effect of sitafloxacin on hepatic function [144]. The authors also performed a study for patients with UTI (*n* = 206) and found similar rates of sitafloxacin AEs for uncomplicated (27.5%) as well as complicated infections (26.5%). Levofloxacin had a lower incidence of AEs in uncomplicated UTIs (21.1%) but similar rates for complicated infections (28.1%) [134]. Numerous other studies further support that the range at which sitafloxacin evokes drug-related AEs varies between approximately 20–30% with hepatotoxicity being the most common that requires monitoring [1,145,174,177]. Nonetheless, considering sitafloxacin belongs to the fluoroquinolones, a class that can potentially elicit life-threatening side effects, its utilization must be warranted by the nature of the patient’s infection.

## 9. Dosing Regimens and Pharmacokinetics

Sitafloxacin has demonstrated efficacy over a range of doses. The appropriate dosage and frequency of administration are selected based on the type of infection, severity of clinical symptoms, and efforts to mitigate potential AE, and medical history. The standard oral tablet of sitafloxacin is 50 mg with the recommended dose for adults being 50 mg bid. Poor response and failure to clear the infection merit increasing dosage to 100 mg bid [1,2,134]. A dosage of 100 mg bid is also warranted for *H. pylori* infection and non-gonococcal urethritis [138,156]; 100 mg qd can be used in the treatment of CAP [178]. Guidelines regarding dosage adjustments for patients who are elderly or have decreased hepatic function remain unclear; nonetheless, drug toxicities may require careful monitoring in these specific patient populations [179]. Sitafloxacin safety in newborns, children, and pregnant individuals has not been established and is therefore not recommended for these patients [179]. In patients with impaired renal function, 50 mg qd is advised for individuals with creatinine clearance (CrCl) between 30–49 mL/min and 50 mg every 48 h for those with a CrCl less than 30 mL/min [1,2]. Critically ill patients who require higher serum concentrations of sitafloxacin may necessitate an IV infusion of 400 mg. However, this is an infrequent occurrence, as the oral form of the drug has good oral bioavailability [145,179].

Investigation of the pharmacokinetics of sitafloxacin has the potential to optimize dosage. For fluoroquinolones, bactericidal efficacy correlates with the 24 h area under the serum concentration–time curve (AUC) to MIC ratio [180]. Specifically, studies in both humans and animal models have demonstrated that an AUC:MIC ratio between 30 and 40 is required for fluoroquinolones (including sitafloxacin) to successfully eradicate bacterial infection [181,182]. Notably, the standard dosage of 50 mg bid for sitafloxacin achieves an AUC:MIC ratio > 100 against 90% of susceptible respiratory pathogens [183]. A different prospective study comparing oral sitafloxacin administration at 100 mg qd to 50 mg bid found higher peak serum concentrations for the 100 mg qd group, suggesting this would be the preferred regimen to treat infection with decreased susceptibility to the antimicrobial. Interestingly, AE rates were increased in the 50 mg bid group (40.4%) versus the 100 mg qd group (33.7%), although the difference was not found to be statistically significant. Nonetheless, the relationship between the dosing schedule of sitafloxacin and AEs deserves further exploration [184].

## 10. Conclusions

Sitafloxacin is able to kill a wide variety of Gram-negative and Gram-positive bacteria. It displays favorable properties such as low MIC for many clinically important pathogens and a good safety profile. Unlike many other fluoroquinolones or antibacterials, it also has potent activity against biofilms. This makes it an interesting option for biofilm-forming infections, such as implant-associated infections. Furthermore, it is able to treat persisters and SCVs, which makes it even more interesting in the fight against recurring infections. Although more studies are needed, sitafloxacin should not only be considered outside of Asia but especially for infections associated with high recurrence coming from persisters or biofilm formation. One such potential future niche is implant- and bone-related infection, as these infections are known to have high recurrence, and often biofilm and persister formation are involved.

## Figures and Tables

**Figure 1 pharmaceuticals-17-01537-f001:**
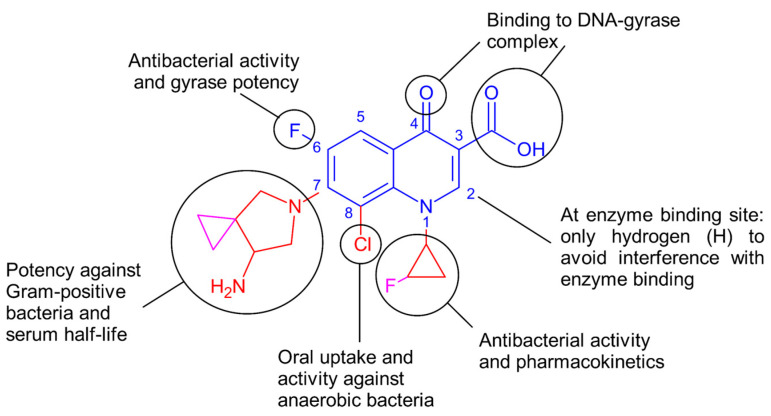
Chemical composition of sitafloxacin with composition-based activity. In blue are the chemical groups typical for all fluoroquinolones. In red are the side-groups that are also present in other fluoroquinolones, and in pink are specific for sitafloxacin. This diagram was adapted from [27,28].

**Figure 2 pharmaceuticals-17-01537-f002:**
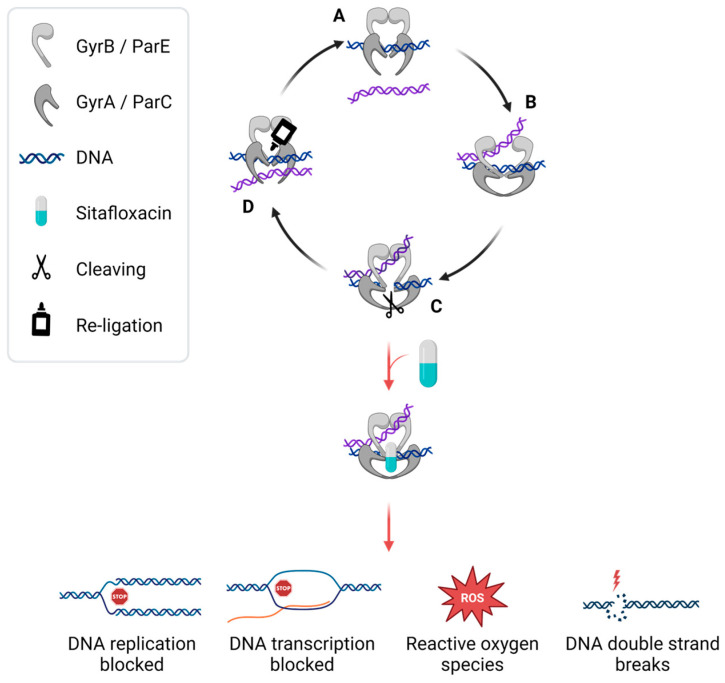
Graphical representation of antibacterial mechanism proposed for sitafloxacin. The regular function of the enzymes DNA gyrase and topoisomerase IV is to first bind one double-stranded (ds) DNA strand (blue strand) (A), which is followed by binding to a second dsDNA (purple strand) (B). Together, these enzymes lead to a controlled break in one of the dsDNA strands (C). The resultant gap allows the second dsDNA to pass through and the re-ligation of the DNA (D). This serves to change DNA topology (relax or supercoil). Gyrase consists of the subunits GyrA and GyrB, topoisomerase IV consists of the subunits ParC and ParE. Fluoroquinolones such as sitafloxacin interfere in this process (at C) by binding to the DNA and both subunits of the enzyme, which stabilizes the cleavage complex. This blocks DNA replication and transcription, leading to the accumulation of reactive oxygen species (ROS) and DNA double-strand breaks, which results in bacterial cell death [27]. Made with BioRender.

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
