# Peer review of "Antibacterial Mechanisms and Clinical Impact of Sitafloxacin"

_pharmaceuticals, 2024, doi:10.3390/ph17111537_

Round 1
Reviewer 1 Report
Comments and Suggestions for Authors
The manuscript needs to be improved:
1. More figures (especially descriptions of antibacterial mechanisms) and comparison tables should be added.
2. Studies on antibacterial activity for the studies highlighted in this review should be highlighted and figures added.
3. How is the ability to resist drug-resistant strains of bacteria? In addition, different factors affecting treatment regimens should be discussed.
Reviewer 2 Report
Comments and Suggestions for Authors
Some of my comments/questions are highlighted by yellow.
This review can be considered for publication only after taking care of its main weak points, listed below according to the order they were provided by the authors.
The authors claim that they discuss the structural characterization, which is an essential tool for the antibiotics activity. However, this info is not provided in this review.
In fact, this manuscript does not provide any structural information. It actually describes only the chemical positioning of specific chemical moieties, hence does not lead to the understanding of the antibacterial mechanisms and the consequent clinical Impact of Sitafloxacin.
Indeed, the authors enlighten the chemical composition, and the positioning of the various moieties of the compound, but not its 3D structure. Thus, the structures of all compounds are determined by single crystal X-Ray diffraction, used in a method called crystallography or by cryo EM. Hence, the authors should replace the word structure by chemical-positioning throughout, including in their concluding summary sentences, and shed light of all expected and obtained properties by the addition of the various chemical moieties discussed in this manuscript.
Examples: The aim of this review is to summarize the chemical-positioning-based mechanisms, activity, resistance profile, and likely future clinical potential of sitafloxacin. In fact, all of the words ‘chemical structure’ should be replaced by chemical composition or chemical positioning
Some specific points/questions
1. It is approved and used in Japan why only in Japan?
2. For only some diseases Why only for these diseases?
3. The ms. says: however, it is also used for ……where? And also: Is it used in ….. illegally?
4. The ms. says: The distinction between the different generations is not clearly defined or clinically important. Prominent examples of these latter generation…. are levofloxacin and moxifloxacin) [1].
Can such generalization be made based of only 2 examples?
Reviewer 3 Report
Comments and Suggestions for Authors
Please see the report attached as pdf file.
